

# A longitudinal study of independent scholar-published open access journals

Bo-Christer Björk, Cenyu Shen and Mikael Laakso

Information Systems Science, Department of Management and Organisation, Hanken School of Economics, Helsinki, Finland

## ABSTRACT

Open Access (OA) is nowadays increasingly being used as a business model for the publishing of scholarly peer reviewed journals, both by specialized OA publishing companies and major, predominantly subscription-based publishers. However, in the early days of the web OA journals were mainly founded by independent academics, who were dissatisfied with the predominant print and subscription paradigm and wanted to test the opportunities offered by the new medium. There is still an on-going debate about how OA journals should be operated, and the volunteer model used by many such 'indie' journals has been proposed as a viable alternative to the model adopted by big professional publishers where publishing activities are funded by authors paying expensive article processing charges (APCs). Our longitudinal quantitative study of 250 'indie' OA journals founded prior to 2002, showed that 51% of these journals were still in operation in 2014 and that the median number of articles published per year had risen from 11 to 18 among the survivors. Of these surviving journals, only 8% had started collecting APCs. A more detailed qualitative case study of five such journals provided insights into how such journals have tried to ensure the continuity and longevity of operations.

## INTRODUCTION

### Background

Individual scientists or groups of scientists were the first to take advantage of the Internet and the web for dramatically re-engineering the publishing of scholarly peer reviewed journals in creating Open Access (OA) journals. Commercial publishers or scientific societies, who have dominated traditional subscription-based publishing of academic journals, have followed much later. In the mid 1990s, electronic-only publishing in conjunction with the OA model seemed ideologically right and suddenly the threshold for starting a journal had dramatically lowered. All that was needed was some server space at the university of one of the editors, someone who mastered a bit of web programming and an enthusiastic group of academics to spread the word via email and academic conferences.

Most of the OA journals founded in the 1990s were of this variety, later many established subscription journals (particularly society ones) have made their digital versions freely available immediately or with a delay. This has been particularly noticeable in countries where cheap or free national or regional electronic portals have become available, like Scielo, Redalyc, and J-stage. Since around 2003 the OA market has become increasingly

Corresponding author
Cenyu Shen, cenyu.shen@hanken.fi

dominated by professionally published journals, which finance themselves by charging authors so-called article processing charges, APCs. At first such journals were being launched by open access publishers like BioMedCentral and PLOS, but in the last couple of years the major commercial and society publishers have increasingly started new OA journals and have also converted some subscription journals to APC-financed models.

Over the years a debate has been raging about the sustainable expenditure for publishing scholarly journals, an issue of particular importance if all journals were to convert to OA (in particular financed by APCs). At one end of the spectrum are the major commercial publishers who claim expenses of around 3,000–5,000 USD per published article (*Morris, 2005*). At the other extreme are scholars engaged in the OA movement (i.e., *Odlyzko, 1997*), who propose that journals can be operated on very low budgets and can be "gratis" at both ends, given that academics can perform almost all needed functions as part of their academic duties anyway, without extra monetary compensation. Much of the publisher-led discussion has been focused on the expenses of IT infrastructure, and copy editing, which are visible parts of the work done in publishing. Less emphasis has been on the tasks involved in coordinating and motivating the network of editors, editorial board members, reviewers, submitting editors etc. which are an essential part of running a journal.

Often the enthusiasm of the founders and their personal network can carry a volunteer-based journal for a few years. But at that same time this type of journal, which lack the support of employed staff and a professional publishing organization, are threatened by many dangers. The editor may change affiliation or retire, or the support of the university hosting the journal might be withdrawn. Authors may stop sending in good manuscripts and it may become more and more difficult to find motivated reviewers. Not being included in the Web of Science, and the impact factor that follows, may in the long run limit the number of submissions severely. On the positive side of the balance the emergence of open source software for publishing (i.e., Open Journals System) and cheap or free hosting services like Latin American Scielo have facilitated the technical parts of publishing.

Now that 15–25 years have passed since the first wave of independent, scholar-published journals were founded, there should be enough concrete evidence to answer questions relating to their sustainability. Much of the data is available freely on the web and it is possible to study which of the early journals have succeeded, which have ceased publishing and which have converted to APC-funding or have been taken over by professional publishers.

## Earlier studies

A number of previous studies, both snapshots and some with longitudinal elements, have shed light on different aspects of such type of journals, which for short we will call "indie" journals.

*Hitchcock, Carr & Hall (1996)* studied electronic English language STM journals available in September 1995. 44 of the 83 relevant journals that they found first appeared in 1995. The share of OA journals varied strongly depending on the type of publisher, with 27% for commercial publishers, 52% for scientific societies and 96% for others (mainly university departments).

*Harter & Kim (1997)* identified 131 electronic journals, which were active in 1996, of which 77 were judged to be peer reviewed journals (of these 39 had published articles in 1993 or before). Since their study was based on e-journal lists compiled in the US, the study had a strong bias to English-language journals. The hard sciences, social sciences and humanities had about equal shares, with the most popular topics being education, literature, mathematics and library and information science. A total of 69% of the journals had only an electronic version while the rest were published in parallel in print. A total of 88% of the journals were OA.

The MSc thesis of *Wells (1999)* was the first to explicitly focus on scholarly or peer reviewed journals, which were free to read (the term Open Access came into use only around 2002). She was able to identify 387 such journals. Probably due to the journal lists she had used as a starting point, over half the journals were published in the US. Overall the vast majority of journals (around 90%) had started publishing in 1994–1998. Her statistics about the organization/person responsible for publishing the journal was interesting: 56% academic, 14% learned society, 13% commercial and 17% other types. For the electronic only journals, 37% were in the social sciences, 20% in life sciences, 19% in arts and humanities, 14% in the physical sciences and 9% in engineering. The overall mortality rate of journals (where the website could not be found or which had not published articles in 1998–1999) was 25%. The highest mortality rates were observed in journals within social science (43%) and humanities (21%).

The first study to look more explicitly at the fate of early OA journals was *Crawford (2002)*. He grouped 104 journals that had been listed in the Association of Research Libraries' (ARLs') Directory of Electronic Journals, Newsletters and Academic Discussion Lists for 1995 depending on their publication output between 1993 and 2000. He found that 27% were publishing substantial amounts of articles, 20% still published small but steady flows of articles, and that 19% seemed "to have fallen prey to the arc of enthusiasm: after a few good years, the journals had died or become comatose." For the rest of the journals their websites were confusing or could not be found.

Our own research group studied OA journals in 2002–2003, using in particular *Wells (1999)* study as a major input. In the first phase of the study (*Gustafsson, 2002*) 317 OA scholarly journals were identified. *Gustafsson (2002)* also studied the status of the journals identified as active in 1998 by *Wells (1999)* and found that 50% were still active in 2002. In the follow-up study (*Hedlund, Gustafson & Björk, 2004*) the editors of all the 317 journals (for which an email address was found) were sent a web survey, for which the response rate was 20%, hence 60 editors answered more detailed questions about their journals. The range of the number of articles published by these 60 journals in 2002 was 3-111, with an average of 20 and a median of 17. The average rate of acceptance for submissions was 50% and 6 out of 60 journals were indexed by the Web of Science. The cost structure of publishing the journals was asked in the form of the time allocation for general tasks such as management and IT-infrastructure (250 h per year) and for the processing of the average article, which was 22 h.

In their conclusions *Hedlund, Gustafson & Björk (2004)* ask the question "The key question for OA publishing is whether it can be scaled up from a single journal publishing

model with relatively few articles published per year to a comprehensive major journal with of the order of 50–100 articles annually.'' They further note: ''The continuation of the journal relies very heavily on the personal involvement of the editor and is as such a risk to the model. Employing staff to handle, for example, management, layout and copyediting tasks, is a cost-increasing factor that also is a threat to the model.'' Both questions are still highly relevant today.

Since the above studies were carried out, the most significant development affecting the publishing of independent scholar-published journals, has been that free or extremely cheap IT-platforms for publishing journals have become available. In particular the Open source software Open Journals Systems has rapidly become very popular as the basic platform both for publishing articles and for managing the peer review process. OJS is currently used by more than 8,200 journals (*PKP, 2016*). *Edgar & Willinsky (2010)* surveyed 3,000 journals using OJS in 2009 and obtained answers from 998 journal editors. The vast majority of the journals charged neither publication fees nor fees for access, and 83% were OA. Academic departments (51%) and scholarly societies (32%) dominated the picture. The geographic spread included South America with 28%, Asia with 13% and Africa 7%. Topically the STM sciences had 40%, social sciences 30% and humanities 11%. The average number of articles published was 31 per year with 74% publishing 0–30 articles, and 9% 60 or more. The study also contains interesting data about the workload done, revenues etc.

Since 2009, the OA journal scene has changed considerably with the increased presence of APC funded commercial publishers. The publishers who are members of the Open Access Scholarly Publishers Association (OASPA) published around 140,000 articles in 2014 (*Redhead, 2015*). Increasingly leading commercial and society publishers are starting new full OA journals or converting existing subscription journals to OA funded by APCs. Unfortunately, academics are also swamped with requests to submit to so-called predatory OA journals with deficient or non-existent peer review practices (*Shen & Björk, 2015*). Much of the debate about Gold OA concerns APC journals and the majority of OA journals which are free to publish in create much less debate, perhaps because many of them are published in languages other than English and in countries outside the US/UK.

Offering qualitative insight into the current challenges of small independent scholar-led journals, *Morrison (2016)* recently interviewed 15 individuals currently involved in producing such journals. Though most believed that the journals would be able to survive in the increasingly competitive OA landscape, many also expressed concerns about their abilities to thrive with existing reliance on external subsidy funding and the level of technical support currently available to them.

The question remains, how sustainable has the independent scholar-published OA journal model been for journals who adopted the model early? The major advantage of these journals some 20 years ago as well as today is that they are rarely based on APC-funding which might be an obstacle for some potential authors without means or mechanisms to fund them.

## METHODS

So far a lot of the discussions (for instance in e-mail discussion lists) about the viability of the 'indie' model has been rather speculative and based on the reporting of anecdotal success stories. Now that over 20 years have passed since the first proper wave of journals founded based on an OA publishing model, there is sufficient longitudinal data available to evaluate the sustainability of the OA model for these early independent OA journals.

The study consists of a quantitative and a qualitative part. The quantitative part builds upon OA journal listings from early OA studies of *Wells (1999)*, *Crawford (2002)*, and *Hedlund, Gustafson & Björk (2004)*. The list of journals from these studies were aggregated into a master list of 264 journals, after which non-independent (e.g., commercially-published, society-published) and journals which had converted from print to OA were removed from the sample (14 in total), resulting in a population of 250 'indie' journals in our study. Each of the journal websites were visited during the second half of 2015 in order to record longitudinal published article volumes, focusing on collecting data about peer-reviewed articles and leaving out editorials and other non-core contents. Some of the earlier studies had already collected and published article volumes and in such cases that data was used and extended. Where available Scopus or DOAJ was used to collect bibliometric information for journals, however, for most the collection was handled manually. It is a cohort type of study, concentrating on journals founded as OA journals prior to 2002 and which we knew were active in that cut-off year. Due to the limited number of journals included in the population sampling was not needed, the whole population was observed.

In the qualitative part the development of five successful 'indie' journals over the past twenty years are described as a multiple case study. The focus is on how the work has been organized and on possible changes in strategies during this time.

## RESULTS AND DISCUSSION

### Quantitative results

In all of the reporting below, 'indie' OA journals which were no longer active and for which no information could be found on the Internet due to completely vanished presence were excluded. When counting the number of OA articles published every year, only articles from journals which have remained with the OA publishing model throughout their lifetime were taken into consideration. Due to a very low number of both 'indie' journals and OA articles in them between 1987 and 1994, the results for journal volumes and article outputs are presented from the year 1995 onwards. For the average OA article per journal, we provided the medians instead of the means due to a highly skewed distribution in the original data, where a couple of very high-volume journals raise the arithmetic mean disproportionately.

### Number of active 'indie' journals over time

Descriptive statistics for the total population of 250 'indie' OA journal founded before 2002 are shown in Table 1. 23 of the journals in the population were categorized as disappeared journals because we could not find their complete information for previous years, they had vanished without a trace and nor was a record of them available from the Internet

**Table 1   Descriptive statistics for the journals included in the study.**

| Population: 250 'indie' journals | | |
|---|---|---|
| Number of active journals | Journals that become subscription journals | 12 |
| | Journals that remain with the OA model | 115 |
| Number of ceased journals | 100 | |
| Number of disappeared journals | 23 | |

Archive. As such they can be considered ceased but there is no way of determining when they stopped publishing so they are treated separately in the results. The rate of still active journals of the 250 journals that we studied was 50.8%, meaning that approximately half of them were still publishing.

Among the still active journals, the majority have remained with the OA publishing model, however a small number of journals (9%) have converted to the subscription-based model. Most of these had been taken over by large commercial publishers using the traditional subscription-based publishing model, on average 12 years after they had been launched. The share of 250 'indie' journals that are currently included in DOAJ and Scopus, are 39.6% and 40.5% respectively. The reason why Scopus has a slightly higher inclusion rate than DOAJ is likely due to the fact that DOAJ has had a policy of removing non-active journals, while Scopus retains historical records of journals regardless of their status. The results also support the fact that nearly half of 'indie' journals have gained good reputation already, since their inclusion in these major indexes and is consistent with the result that about half of the journals were still active as at 2014.

The development over time of active 'indie' OA journals before and after 2002 is shown in Figs. 1A and 1B. A journal was counted as 'active' in a particular year if it was still publishing articles in that year. Before 2002 the number of active journals grew very rapidly from a total of 76 journals in 1995 to 207 journals in 2002. The year 2002 was the cut-off year to be included in the studied cohort, meaning that no new journals were added to the data set after this point in time. After 2002, the number of journals in the cohort decreased steadily to the 127 that stayed active in 2014.

The share of surviving journals currently charging APCs was also observed, with a result that nearly 8% among them did so. Clearly this is a viable option for keeping a journal running, provided that the submission levels do not drop significantly after this. It is also an option which is better suited for journals in the hard sciences, in particular health and biomedicine journals, due to the fact that APC-funded OA journals are more common with funding mechanisms more widely established and available to researchers.

### Annual median number of articles per journal

Figures 2A and 2B describe the annual median number of articles published by the 'indie' journals. The number was relatively stable at around 10 articles in the period 1995–2002. This may be the result of many of the journals just having started their existence at that time, with understandably few articles in the first few issues. After 2002, the median for active journals increased steadily from 11 in 2003 to 18 in 2014. Again, this is probably a result of the fact that the surviving journals had already established a good reputation and increased submission numbers during the period, i.e., the stronger getting stronger.

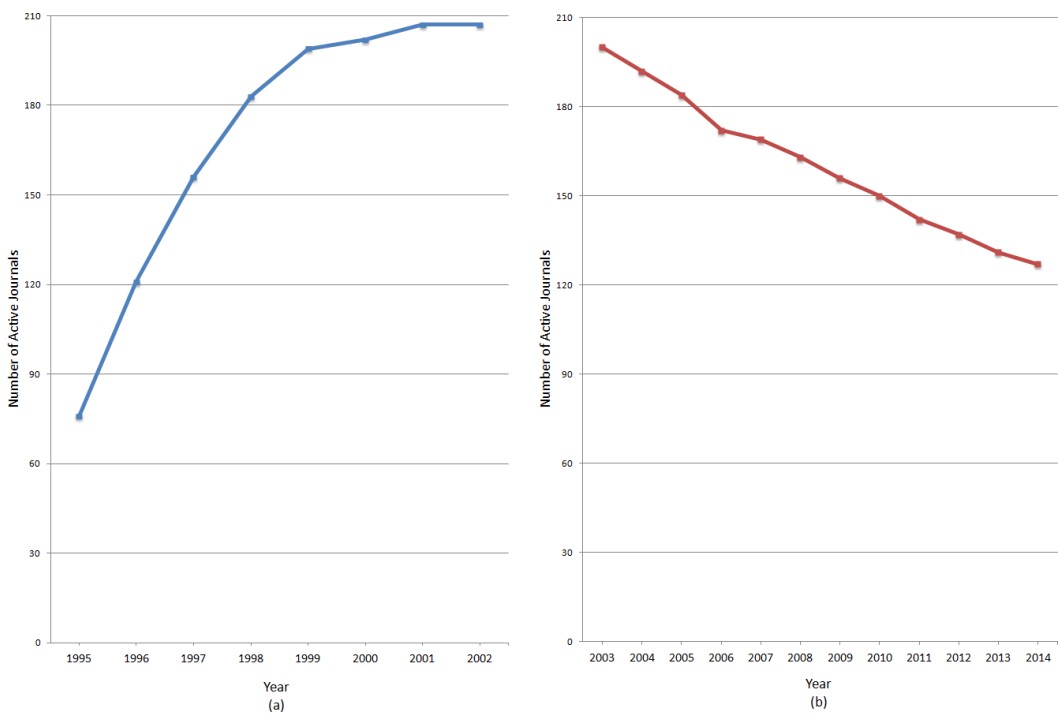

**Figure 1** **The number of active 'indie' journals from the cohort of journals founded prior to 2002.** (A) Results between 1995 and 2002; (B) Results after no new journals were added from 2003 till 2014.

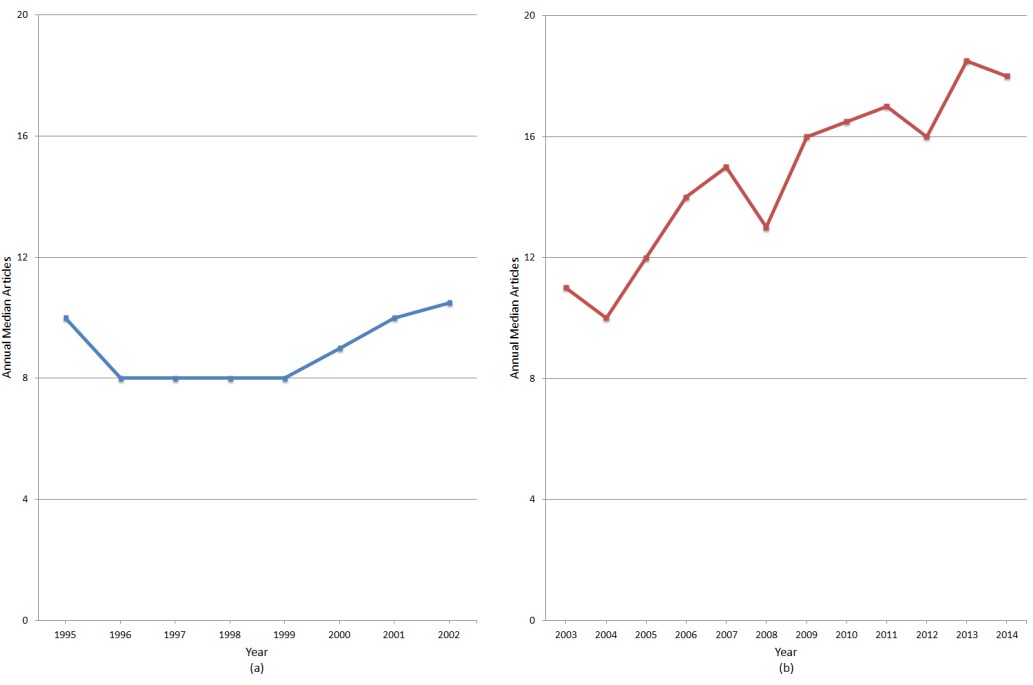

**Figure 2** **Annual median number of articles published by 'indie' journals from the cohort of journals founded prior to 2002.** (A) Results between 1995 and 2002; (B) Results after no new journals were added from 2003 till 2014.

**Table 2** Main results datasheet.

| Year | Annual median articles per journal | Total number of articles | Number of active journals | Number of ceased journals |
|------|-----------------------------------|--------------------------|---------------------------|----------------------------|
| 1995 | 10 | 862 | 76 | 1 |
| 1996 | 8 | 1,305 | 121 | 2 |
| 1997 | 8 | 1,819 | 156 | 3 |
| 1998 | 8 | 2,243 | 183 | 1 |
| 1999 | 8 | 2,593 | 199 | 2 |
| 2000 | 9 | 3,095 | 202 | 4 |
| 2001 | 10 | 3,135 | 207 | 1 |
| 2002 | 11 | 3,563 | 207 | 5 |
| 2003 | 11 | 3,715 | 200 | 7 |
| 2004 | 10 | 3,497 | 192 | 8 |
| 2005 | 12 | 4,241 | 184 | 8 |
| 2006 | 14 | 4,090 | 172 | 12 |
| 2007 | 15 | 4,394 | 169 | 3 |
| 2008 | 13 | 4,760 | 163 | 6 |
| 2009 | 16 | 4,681 | 156 | 7 |
| 2010 | 17 | 5,083 | 150 | 6 |
| 2011 | 17 | 5,290 | 142 | 8 |
| 2012 | 16 | 4,934 | 137 | 5 |
| 2013 | 19 | 5,298 | 131 | 6 |
| 2014 | 18 | 4,954 | 127 | 4 |

The detailed results are included in Table 2 together with other main results for total article volumes. There is a continuous growth in the number of articles published by active journals over the past decades from 862 articles in 1995 to 4,954 articles in 2014, despite the fact that no new journals were added after 2002 (when 3,563 articles were published). The growth between 2003 and 2014 is explained by the growth of the per journal publishing figures. These findings suggest that at least half of the 'indie' journals are quite sustainable, at least at their current publishing level.

### Age distribution for ceased journals

Figure 3 illustrates at what age ceased 'indie' journals stopped publishing. Most journals survived the first 2–5 years period, whereas the mortality rate rose in the critical 6–9 years period. After that, the number of journals ceasing dropped sharply, indicating that the surviving journals had found stability.

### Subject fields

Figure 4 describes the distribution of total OA articles published across different subject fields. During the four time periods we studied, the article volumes have shown a fast growth in most disciplines except for chemistry, physics and astronomy as well as business and economics. The largest share of articles was in the social sciences with nearly 8,000 articles published in the most recent four years, followed by mathematics with almost 6,000 articles. The discipline of earth science and biomedicine ranked third (4,123 articles) and fourth (3,383 articles).

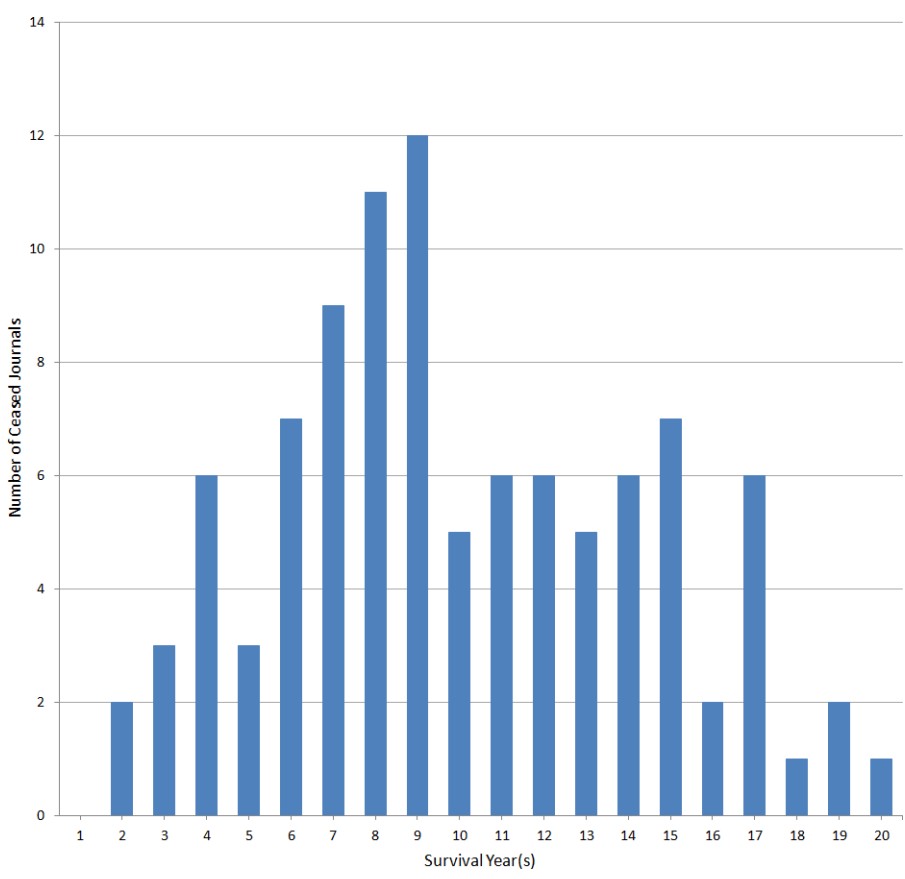

**Figure 3** Age distribution for ceased journals.

## Five cases

In the following five case journals, all founded around 1995 are presented. All journals are still surviving and they were picked as a convenience sample, representing slightly differing evolutionary paths. Two of the journals are nowadays charging APCs and one of these has been turned over to a professional OA publishing company. For some of the discussed journals there are published case studies and descriptions available. One of the authors of this article was the founder and long-time editor of ITcon. As for MEO an interview was conducted with the former editor-in-chief.

### Electronic Journal of Information Technology in Construction

The journal, abbreviated ITcon, was founded by four researchers from different parts of the world, active in the same research area, and part of a network meeting at regular yearly conferences (*Björk & Turk, 2006*). The first author of this study was the editor-in-chief, and hence the official publisher was at first his university at the time being. Later when he moved to another university an international organization of building researchers was asked to be the official publisher, but this has in fact been a "rubber stamp." The web server used has all the time been at the university of another one of the co-editors, who also programmed the first software needed to publish articles. The software has undergone

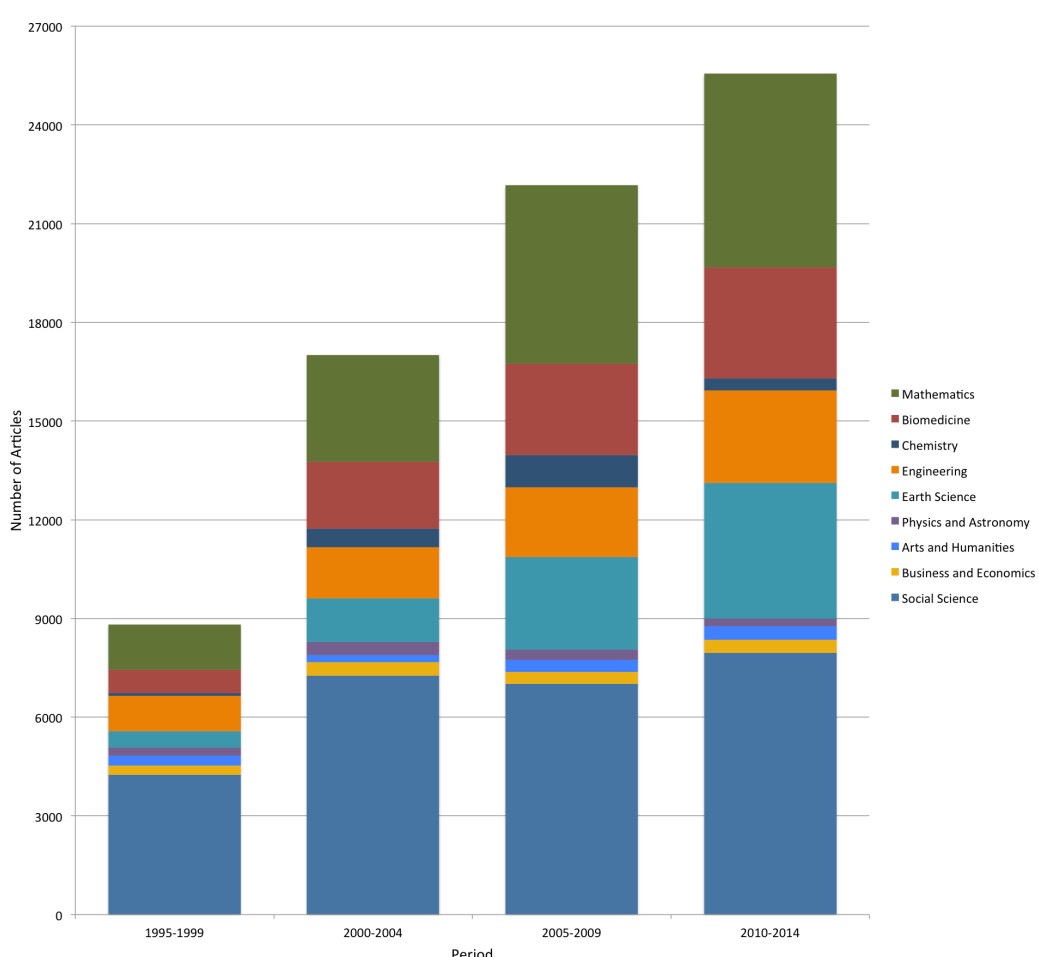

**Figure 4** The distribution of OA article volumes over time and across subject fields.

a couple of revisions, partly indirectly funded via an EU research grant, and also includes facilities for managing submissions and reviews. That part has however been difficult to use and is no longer actively used. There have been discussions about going over to the use of OJS or some similar open source software, but so far this has not been done.

The first four years the journal was struggling to get submissions and only published 3–5 articles per year. It was saved by the suggestion of one of the editorial board members to start publishing special issues, which previously had been deemed out of the question since the aim was to publish papers as soon as they had gone through the review process. Since 2001 the journal has regularly published more than half of its articles in special issues. The experience is, that researchers are more keen to publish in special issues than to submit individual papers. Also it is usually easy to outsource the whole process to volunteers acting as guest editors for an issue.

Looking at the competition of 3–4 subscription journals with similar topics, ITcon would really have benefitted from inclusion in Web of Science and getting an impact factor. One attempt was made, in 2006, but unsuccessfully. Currently the journal is "struggling" with a yearly output of between 20 and 40 articles. What probably has saved the journal (which

has no income but is fully based on volunteer work and the access to a free server) is the tiered managerial structure of up to ten co- and junior editors, which helps in spreading the workload of overseeing the review of submitted manuscripts. It can thus be characterized as a collective endeavor.

### Journal of Medical Education Online

The idea of the Journal of Medical Education Online, abbreviated MEO, came from the founding editor's dissatisfaction around 1995 with how traditional academic publishers were ignoring the new opportunities offered by the Internet. After discussions with several colleagues, David Solomon originally envisaged MEO as a forum for knowledge exchange for both researchers and educators in the field, with a peer-reviewed journal as the center-piece (*Solomon, 2007*). Over the years the additional features did not really catch on, but the peer-reviewed section has remained. After a few struggling years with below ten published articles per year, the journal started to rapidly gain momentum reaching 100 yearly submissions in 2006.

Similar to many other journals with a heavy involvement of the editor-in-chief MEO has changed publishing venue according to the editor's affiliation, depending on his employer allowing a considerable input of work time. In the case of MEO, the editor in addition to overseeing the reviewing spent a lot of time on developing both the publishing and workflow platform during the first ten years of the journals existence. In 2005, the rapidly grown submission flow necessitated a restructuring of the editorial process so that a number of co-editors helped share the burden. There were also plans to move the journal back to the original university in Texas and for new people to take over the responsibility of the journal, but these plans were partly upset by hurricane Katrina. Instead, one of the people, Ann Frye joined David Solomon in co-editing the journal.

An important goal for the journal has all along been indexing in major indexes, including Medline. A prerequisite for getting accepted in that index was the formatting of articles in XML meeting the National Library of Medicine's (NLM) standards which requires a great deal of expertise. The journal began charging an APC of 100 USD to cover the expense of using a professional service to format articles and create an XML version meeting NLM standards. The fee was raised to 200 USD a year later to cover professional copy-editing. Getting accepted in Web of Science took an additional three years after the journal was accepted into Medline and Scopus.

Around 2009, the two editors decided they wanted to use a professional publisher freeing them to focus on the editorial tasks and providing professional level publishing services. Since an APC of 200 USD had not reduced submissions, they felt it was worth the risk of increasing the APC considerably to have the journal professionally published. Discussions were started with a small OA publisher, Co-Action publishing. David Solomon knew and trusted the owners of the company have worked with one of the principals of the company in forming OASPA. Initially, the two editors retained 50% ownership with Co-Action receiving 50% ownership for taking full financial responsibility for the journal. After two more years, the original editors decided to step down as editors and gave full ownership to Co-Action and engaged new editors who receive a small fee for the work

from the publisher. After an initial APC of 600 USD the APC has been increased in several increments to 1500 USD, while the number of publications per year has steadily increased.

### Information Research

The origins of IR were in a newsletter from the centre of user studies at the department of information studies at the university of Sheffield, which from 1990 was transformed into Information Research News, which published working papers in print (*Wilson, 1998*). From April 1995 a parallel electronic version was published and in 1997 the print version was ended. Initially the focus was on publishing un-refereed papers from the department, but gradually the focus shifted to peer-reviewed papers with mainly outside authors. The university provided indirect support in the form of the web server and allowing staff the time to work with the journal.

Later as the editor retired the journal has shifted locus. The journal home page currently says about the journal: "It is privately published by Professor TD Wilson, Professor Emeritus of the University of Sheffield, with in-kind support from Lund University Libraries, Lund, Sweden and from the Swedish School of Library and Information Science." The home pages also contain a plea for sponsorship either directly in money or in kind, as well as advertisements. In particular, the journal is seeking volunteers to help in copy-editing or formatting. In a recent web survey of readers or authors (*Wilson, 2012*) the respondents were directly asked how likely they would be to continue submitting articles to the journal if it had to start collecting APCs.

The journal has a stable output of four issues a year. The journal also contains book reviews, conferences announcements etc. It is highly ranked in the rankings of information system journals in many countries.

In contrast to many other 'indie' journals, Information Research has remained with html for the published papers, assuming readers would read directly from the screen. This format enables hyperlinking references in the text with the list of references.

The key to success for IR seems to have been in quite rapidly being able to publish a full journal with quarterly issues, at a time when major publishers were just starting to publish electronic versions of their paper journals. Probably early indexing in Web of Science, with a resulting impact factor, has also helped the success of the journal.

Despite having its origins in a print departmental newsletter, we feel that Information Research can be characterized as an 'indie' journal, especially after severing the ties with mother organization.

### Electronic journal of Geotechnical Engineering

This is an interesting case of an 'indie' journal turned predatory. Published by a now retired professor from an American university, the first issue in 1996 contained invited papers after which the journal was a typical struggling 'indie' journal with a slowly rising publication volume from 4 to 33 papers between 1997 and 2007. After that the volume has dramatically risen to 628 in 2014. Jeffrey Beall wrote a blog accusing the journal for having turned predatory in July 2015 (*Beall, 2015*). Currently the journal pages say "editorial fee is $500 for the entire editorial and publishing work. Following the "supply and demand" rule of economics, this may be modified". The journal website still has an amateurish 1990's

feel and look (authors are instructed that they can also send the files on floppy disks!) and authors sign over the copyright to the journal.

### First Monday

The first publisher of First Monday was curiously a Danish commercial publisher (Munksgaard) that was keen on experimenting with the new medium that the Internet offered. In 1999 the journal was bought by the editor Edward Valauskas, whose idea the journal had been from the start, with two colleagues (*Pauli, 2011*). Valauskas had all the time insisted that the journal be OA and that authors retain copyright, whereas Munksgaard had planned for the journal to evolve into a subscription journal after an initial open offering. Since then the journal has been hosted by the University of Illinois in Chicago and is currently using the OJS platform.

First Monday is published on the first Monday of each month, a reference back to the schedule of the first scholarly journal: Philosophical Transactions of the Royal Society. For a journal with no monetary budget it is quite amazing that it has published more than 1,500 articles, and despite the fact that the journal is not indexed in the Web of Science. One of the reasons for the popularity of the journal is that it is a multidisciplinary journal about the phenomenon Internet and that it has achieved a strong following in that niche area.

In 2011 the acceptance rate was as low as 15%, which means that there is a lot of work behind each published article.

## CONCLUSIONS

The 'indie' journals that comprise this study are children of their time, early pioneers in adopting a disruptive innovation for scholarship. With no or very little subscription income, and author-fees being an unestablished concept, the circumstances for running an independent journal was certainly a challenge in the 1990s and early 2000s. This study is limited to observing the sustainability trajectories of early 'indie' journals, the thousands of similar journals founded since then might have very different characteristics and warrant focus in future studies.

Nowadays the overall scene for launching OA journals looks very different than it did some 20 years ago. Currently there are a few specialized OA publishers and also the big established subscription publishers. The norm nowadays is also increasingly to collect APCs. Nevertheless early 'indie' journals have played an extremely important role in promoting OA, and some 'indie' journals have become important journals in their niche areas. Comparing the longitudinal publishing and citation metric trajectories of 'indie' journals to commercially-operated counterparts could also be a potential avenue for future studies to explore.

The quantitative study shows that even successful 'indie' journals tend to be rather small, the median being 18 articles per year. A fairly large share of articles is in the social sciences and humanities. Areas such as mathematics and earth sciences have, however, grown in importance over the years. Looking at the mortality rate of journals, it is evident that the years 6–9 are crucial. The initial enthusiasm can sustain even a very low volume journal for a while, but after the journal has to have a reasonably good inflow of manuscripts, both

to be able to ensure the quality of published articles as well as credible article numbers. We also already found that 8% of the surviving 'indie' journals have started collecting APCs as a means to ensure enough revenue to keep the journal running.

The cases studies demonstrate that remaining pioneer journals could consider exploring alternative scenarios in order to ensure survival. Journals operating in the hard sciences may well start charging APCs, given that mechanisms for funding these are evolving in many countries. An important issue for all such journals lacking the backing of a professional publishing organization, old or new, is to provide for a generation shift in the editor function.

### Funding
The authors received no funding for this work.

### Competing Interests
The authors declare there are no competing interests.

### Author Contributions
- Bo-Christer Björk conceived and designed the experiments, performed the experiments, analyzed the data, wrote the paper, reviewed drafts of the paper.
- Cenyu Shen performed the experiments, analyzed the data, wrote the paper, prepared figures and/or tables, reviewed drafts of the paper.
- Mikael Laakso conceived and designed the experiments, performed the experiments, wrote the paper, reviewed drafts of the paper.

### Data Availability
   The raw data has been supplied as Supplemental Information.

### Supplemental Information
Supplemental information for this article can be found online at http://dx.doi.org/10.7717/peerj.1990#supplemental-information.

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
