# Peer review of "A longitudinal study of independent scholar-published open access journals"

_PeerJ, doi:10.7717/peerj.1990_

## Round 0.1 · original submission · Minor Revisions

Please think about the title. You'll see that both reviewers commented on whether the current title is too broad so I recommend you narrow the focus and perhaps add some more details of the study scope. I hope you'll also consider the reviewers' other comments, especially on clarifying the figures.

·

Basic reporting

No comments

Experimental design

No comments

Validity of the findings

No comments

Additional comments

This study establishes new knowledge that will be important in our understanding of the options available to the scholarly community when shaping the future of scholarly communication. It is well conceived and conducted, and merits publishing without substantial changes.

Some general comments for the authors:
Lines 196–199: “Now that over 20 years have passed there is enough evidence out there openly on the web to be able to provide empirical data on how sustainable the scholar-published journal model is. This is the research question for this study.”

a) It should be said that this study looks at how sustainable this model is for the individual journal, not on how viable the model is as a possible solution for science in general. I think it is important to stress this, as there are many arguing for a transition to this kind of scholar-based publishing as an answer to “all” problems with commercial publishers.

b) The choice of journals studied is a specific subset, which must be assumed to have certain characteristics that is not necessarily shared with newer upstart journals. These were the pioneers, seeing new opportunities in the new technologies. Findings for the pioneers may not necessarily be valid for later entrants. Such a validity is not claimed by the authors, but it could be pointed out more clearly. (A follow-up study on newer indie journals, and their commercial counterparts, could be called for.)

I also have a problem with the perspective of some of the works cited, e.g. Odlyzko (1997) (line 87 ff) where there is an obvious confusion of the term “cost” with “expenditure”. Cost is the use of resources, while expenditure indicates financial outlays. The use of professorial time is a cost, even if it carries no financial outlay if the professor uses his working hours to publish a journal. I contend that indie journals have huge hidden costs. The authors do not, as far as I can see, subscribe to this confusion of the terms, but I would like some comments that makes this clear.

More specific points with minor comments from me:
Line 81: Hindawi started up as a traditional publisher in 1997, so I wonder if it is correct to list it as a start-up in this OA context? (see https://en.wikipedia.org/wiki/Hindawi_Publishing_Corporation)

Lines 136–137: The number 21 % for humanities does not fit well with “much higher” when compared to 25 %

Lines 169–170: PKP on their webpage say 8,286 in 2014. see https://pkp.sfu.ca/ojs/ojs-usage/

Lines 200 ff: It is a bit difficult to follow these numbers, e.g. the “Total” of 264 in line 230. Is this the sum of all journals studied in the works referenced, or is it the number of journals still “alive” when those studies were conducted? A few clarifying words would be appreciated.

Line 242: A bit surprising to find more journals listed in SCOPUS than in DOAJ, I would have thought the latter the easiest to get into. Any indication of major reasons many are not listed in DOAJ?

·

Basic reporting

The article is structured clearly and in a way that maps to the flow of the research. It is written in clear English at a suitable level that an interested generalist would understand it. Given the broad potential audience this is useful.

I feel that the presentation of the figures could be improved. My view is that the line graphs should be presented as column plots as they are categorical (year by year) data. In any cases the lines between points are confusing and misleading, particularly for Figure 3.

Figure 1 is confusing as it really combined two different analyses, the growth of a cohort and then the decline of that cohort. I would present as either two separate charts or clearly delineate the two phases by changing colour of the two phases.

I found the description of the cohort confusing at some stages and what was included and excluded from specific figures. It might be helpful to add a little more detail to the figure legends to be explicit about exactly which parts of the cohort are included in each analysis.

Experimental design

The experimental design is straightforward. The cohort is clearly described (and made available in the supplementary data). The analysis is straightforward and replicable (again, against the provided data) and sufficiently described.

The supplementary data itself is provided as an Excel spread sheet which appears to have dependencies to another file which is not made available. I would recommend severing the dependencies and additionally making the dataset available as a CSV file for those without access to Excel. For best practice it would be useful to include a data dictionary that explicitly describes the column contents and meaning of data elements (e.g. the first column is whether an APC was being charged in 2016 I presume)

For the dataset itself it would be valuable to include (if possible) date of inclusion on WoK and Scopus (where relevant) and additionally the year when APCs were introduced (where applicable). Publisher history would also be valuable but probably involves substantially more work. The rationale behind asking for dates is to support analysis of the correlations between eg indexing in Scopus and article numbers that are suggested from the Case Studies.

Validity of the findings

The finding as presented are valid and consistent with the evidence provided. However it seems to me that the paper doesn't really answer the question posed in the title (on which more below).

Additional comments

The paper is a useful contribution on the history and success of a specific cohort of journals. However it falls somewhat short of the promise of the title in two ways: it doesn't really answer the question of whether this cohort are viable in the long term, and it explicitly notes that the analysis does not provide guidance to those starting indie journals today.

While the paper is publishable as stands with some minor modifications I believe it would substantially strengthen the paper if the suggestions that emerge from the Case Studies were to be tested by a further analysis of the data. This could in turn provide guidance on where scholar-journals should focus their efforts.

There is, for instance no analysis as to how article numbers relate with indexing, or the introduction of APCs. While this could be done independently with the data provided I feel it would strengthen the paper which is otherwise of more historical interest than providing insight into journal development for today.

If the decision is taken not to do this further analysis I would change the title to reflect that it focusses on an analysis of a historical cohort that may not provide much insight into journal survival in 2016.

---

## Round 0.2 · Minor Revisions

Thanks for responding to the reviewers' comments. I read the revised article carefully and suggest just one important correction (to the abstract) and some minor corrections to the language (as PeerJ doesn't do any further copy editing). These are listed on the attached PDF. As they are minor, you don't need to submit a formal rebuttal letter.

---

## Round 0.3 · accepted · Accept

Thanks for your prompt response. I hope you found the minor language corrections helpful!